# Systematic estimates of the global, regional and national under-5 mortality burden attributable to birth defects in 2000–2019: a summary of findings from the 2020 WHO estimates

Jamie Perin [iD],[1] Cara T Mai [iD],[2] Ayesha De Costa [iD],[3] Kathleen Strong [iD],[3] Theresa Diaz [iD],[4] Hannah Blencowe [iD],[5] Robert J Berry [iD],[2] Jennifer L Williams [iD],[2] Li Liu [iD] [6]

For numbered affiliations see end of article.

**Correspondence to**
Dr Jamie Perin; jperin@jhu.edu

## ABSTRACT

**Objectives** To examine the potential for bias in the estimate of under-5 mortality due to birth defects recently produced by the WHO and the Maternal and Child Epidemiology Estimation research group.

**Design** Systematic analysis.

**Methods** We examined the estimated number of under-5 deaths due to birth defects, the birth defect specific under-5 mortality rate, and the per cent of under-5 mortality due to birth defects, by geographic region, national income and under-5 mortality rate for three age groups from 2000 to 2019.

**Results** The under-5 deaths per 1000 live births from birth defects fell from 3.4 (95% uncertainty interval (UI) 3.1–3.8) in 2000 to 2.9 (UI 2.6–3.3) in 2019. The per cent of all under-5 mortality attributable to birth defects increased from 4.6% (UI 4.1%–5.1%) in 2000 to 7.6% (UI 6.9%–8.6%) in 2019. There is significant variability in mortality due to birth defects by national income level. In 2019, the under-5 mortality rate due to birth defects was less in high-income countries than in low-income and middle-income countries, 1.3 (UI 1.2–1.3) and 3.0 (UI 2.8–3.4) per 1000 live births, respectively. These mortality rates correspond to 27.7% (UI 26.6%–28.8%) of all under-5 mortality in high-income countries being due to birth defects, and 7.4% (UI 6.7%–8.2%) in low-income and middle-income countries.

**Conclusions** While the under-5 mortality due to birth defects is declining, the per cent of under-5 mortality attributable to birth defects has increased, with significant variability across regions globally. The estimates in low-income and middle-income countries are likely underestimated due to the nature of the WHO estimates, which are based in part on verbal autopsy studies and should be taken as a minimum estimate. Given these limitations, comprehensive and systematic estimates of the mortality burden due to birth defects are needed to estimate the actual burden.

## INTRODUCTION

Birth defects (or congenital anomalies) are a group of structural or functional conditions

### STRENGTHS AND LIMITATIONS OF THIS STUDY

⇒ Estimates from WHO are compiled and represented systematically for country groups of interest.
⇒ The limitations of the WHO estimates for mortality due to birth defects are thoroughly described and considered in the context of birth defects and the rapid decline of under-5 mortality.
⇒ This study is limited in that estimates of mortality due to birth defects from the WHO are not compared with other similar estimates such as those produced by the Global Burden of Disease study.
⇒ This study did not examine estimated mortality for specific types of birth defect.

that occur in utero. Birth defects can be identified prenatally, at birth or later in life. Causes of these conditions are multifactorial and can incorporate genetic, behavioural and environmental factors, but in many cases, the causes of birth defects are unknown. Birth defects impact affected individuals and their families, including mortality, morbidity, the cost of care and individual and family well-being.[1] As under-5 mortality has declined, birth defects have become a more prominent cause of death in young children.[2] Despite the acknowledged impact and need for monitoring to guide programmatic action, birth defects are not tracked in many health systems.[3] Health systems in low and middle-income countries (LMICs) are not generally equipped with the capacity to screen for or diagnose birth defects.[4] Even in high-income countries with birth defects surveillance systems and high vital registration coverage, the measurement of birth defects is not always consistent between countries or consistent over time.[5] Modelled estimates are, therefore,

required to describe the global burden of birth defects and their contributions to under-5 childhood mortality.

Since the mid-2000s, there have been several efforts to estimate the burden of under-5 mortality due to birth defects by different groups. Modelled estimates for all countries were first developed in 2006 by the March of Dimes in a collaboration with the WHO.[6] More recently, estimates of the under-5 mortality due to birth defects based on the European Surveillance of Congenital Anomalies were developed and have been iteratively adapted for specific settings.[7–9] In addition, the Global Burden of Disease (GBD) study from the Institute of Health Metrics and Evaluation has recently developed estimates for under-5 mortality due to birth defects.[10]

An additional effort to estimate causes of mortality in children under 5, including the mortality due to birth defects among newborns (age 0–27 days) and older children (28 days or 1 month to 59 months), has been ongoing since 2008, led by the Maternal and Child Epidemiology Estimation (MCEE) group in collaboration with the WHO. Its most recent update covers all WHO member states for the years 2000–2019.[2] This series of estimated cause-specific mortality provides much detail about the mortality burden of birth defects globally and for specific countries over time.

We describe here the burden of under-5 mortality due to birth defects as currently estimated by the MCEE and the implications for health policy and planning. We also describe the limitations of these estimates. In light of these limitations, we describe the need for less biased estimates of the burden of birth defects (eg, less sensitive to verbal autopsies) and make a call to action to the public and international health community to produce these estimates.

## METHODS
### Data sources
In order to estimate cause-specific mortality for children under 5, the MCEE uses data from several distinct sources. The first is from vital registration systems, collected and maintained by the WHO. Cause of death data from vital registration covers approximately 40% of countries worldwide. However, vital registration does not generally capture a large portion of all under-5 deaths because countries with vital registration are generally high income and have fewer deaths among children under-5 years.[11] Although many countries contribute vital registration data to the WHO, the MCEE only uses data from those high-quality systems that cover at least 85% of the estimated deaths by age group and have a high percentage of deaths that use well-identified International Classification of Diseases (ICD) coding (eg, that avoid codes for conditions that do not cause mortality). For neonatal mortality, 73 out of 194 countries contribute vital registration data, while 76 contribute vital registration data for children 1–59 months.

The second primary data source for the MCEE estimates of under-5 mortality due to birth defects is based on results from verbal autopsy studies in areas without high-quality vital registration. The MCEE conducted a systematic review of published verbal autopsy studies that reported at least two causes of death for samples representative of the general population in areas with high neonatal and under-5 mortality, from 1980 to 2018. This review identified studies with population representative samples, such as those identified in large household surveys. In general, these deaths did not occur in health facilities or with skilled medical attendance. The cause of these deaths is determined with the best available information, which is obtained from a verbal autopsy interview with family or caregivers,[12] which are then summarised to determine cause of death with an algorithm, several of which are in common use.[13] Verbal autopsy interviews generally include at least one question related to birth defects, such as 'Was any part of the baby physically abnormal at the time of delivery? (eg, body part too large or too small, additional growth on body)',[14] which can be used by a reviewing physician or algorithm to determine mortality due to birth defects.

### Statistical methods
The MCEE research group has published their methodology for estimated cause-specific mortality for children under 5 in detail.[2 15] In brief, vital registration data are used for low mortality countries (defined as less than 10 per 1000 live births for neonates and 25 per 1000 live births for 1 to 59 months). For moderate and high-mortality countries (defined as 10 or more deaths per 1000 live births for neonates and 25 or more per 1000 live births for 1 to 59 months), verbal autopsy data identified by systematic reviews are used to model cause-specific mortality estimates. Cause-specific mortality is modelled separately by mortality level (moderate to high or low) as well as age, conditional on a small set of covariates related to child health that are chosen separately for each age group and mortality strata. These models are estimated in a Bayesian framework, which employs the least absolute shrinkage and selection operator to select relevant covariates without overfitting. MCEE estimates are reviewed by countries in a consultation process where country representatives can ask questions or suggest alternate sources of causes of death.[2]

We summarised the MCEE estimates for deaths due to birth defects as both the number of deaths and the per cent of neonatal, 1–59 month, and under-5 mortality. We also examined the mortality rate per 1000 live births for each of these three age groups by WHO six regions (online supplemental annex 1 WHO Member States by Region—African, Region of the Americas, Eastern Mediterranean, European, South East Asian and Western Pacific), by national income (online supplemental annex 2) and by under-5 mortality rate in 2000, 2010 and 2019.

**Table 1** Estimates in 2000, 2010 and 2019 for deaths due to birth defects among children under 5

| Year | Deaths due to birth defects (thousands), (95% UI) | | Mortality rate (per 1000 live births), (95% UI) | | Per cent of all under-5 deaths, (95% UI) | |
|---|---|---|---|---|---|---|
| Neonates (0–27 days) | | | | | | |
| 2000 | 276.3 | (237.6 to 324.7) | 2.1 | (1.8 to 2.5) | 6.9% | (6.0 to 8.1) |
| 2010 | 262.4 | (226.5 to 303.6) | 1.9 | (1.6 to 2.2) | 8.6% | (7.4 to 9.9) |
| 2019 | 236.7 | (207.9 to 279.5) | 1.7 | (1.5 to 2.0) | 9.7% | (8.5 to 11.5) |
| 1–59 months | | | | | | |
| 2000 | 176.2 | (162.5 to 190.2) | 1.3 | (1.2 to 1.4) | 3.0% | (2.8 to 3.2) |
| 2010 | 175.8 | (157.4 to 185.0) | 1.3 | (1.1 to 1.3) | 4.5% | (4.1 to 4.8) |
| 2019 | 167.2 | (145.8 to 181.0) | 1.2 | (1.0 to 1.3) | 5.9% | (5.1 to 6.3) |
| 0–59 months | | | | | | |
| 2000 | 452.6 | (408.9 to 502.5) | 3.4 | (3.1 to 3.8) | 4.6% | (4.1 to 5.1) |
| 2010 | 438.2 | (392.7 to 477.6) | 3.2 | (2.8 to 3.5) | 6.3% | (5.7 to 6.9) |
| 2019 | 404.0 | (367.6 to 455.7) | 2.9 | (2.6 to 3.3) | 7.6% | (6.9 to 8.6) |

.UI, uncertainty interval.

## Patient and public involvement

There were no patients involved in this study, and results were not disseminated to patients or patient families. All estimates and source data are publicly available at https://github.com/amulick/MCEE-u5mort2019.

## RESULTS

In 2000, under-5 deaths due to birth defects across all countries were estimated to be 452 600 (95% uncertainty interval (UI) 408 900–502 500). In 2019, under-5 deaths due to birth defects dropped to 404 000 (95% UI 367 600–455 700). The under-5 deaths per 1000 live births from birth defects fell from 3.4 (95% UI 3.1–3.8) in 2000 to 2.9 (95% UI 2.6–3.3) in 2019. Conversely, the per cent of all under-5 deaths attributable to birth defects increased from 4.6% (95% UI 4.1%–5.1%) in 2000 to 7.6% (95% UI 6.9%–8.6%) in 2019 (table 1). Across all

time periods, mortality due to birth defects was higher among neonates than in older children.

Estimates for deaths due to birth defects among children under-5 by high-income countries and LMICs are shown in table 2. LMICs disproportionally shouldered the burden of total global deaths over time, with 19–27 times more deaths due to birth defects reported in 2000, 2010 and 2019 than in high-income countries. Rates per 1000 live births dropped in high-income counties from 2.00 (95% UI 1.97–2.03) in 2000 to 1.29 (95% UI 1.24–1.34) in 2019, while rates in LMICs remained relatively stable, 3.58 (95% UI 3.21–4.00) in 2000 and 3.03 (95% UI 2.75–3.36) in 2019. The per cent of deaths attributable to birth defects in high-income countries has remained stable over the 20 years examined (~28%), while the per cent of deaths attributable to birth defects in LMICs has increased (4.4% (95% UI 3.9%–4.9%) in 2000 to 7.4% (95% UI 6.7%–8.2%) in 2019).

**Table 2** Estimates in 2000, 2010 and 2019 for deaths due to birth defects among children under 5 for high-income and low-income and middle-income countries

| Year | Deaths due to birth defects (thousands), (95% UI) | | Mortality rate (per 1000 live births), (95% UI) | | Per cent of all under-5 deaths, (95% UI) | |
|---|---|---|---|---|---|---|
| High-income countries | | | | | | |
| 2000 | 22.8 | (22.5–23.2) | 2.0 | (2.0–2.0) | 28.8% | (28.4–29.3) |
| 2010 | 18.0 | (17.7–18.3) | 1.5 | (1.5–1.6) | 28.3% | (27.9–28.8) |
| 2019 | 14.3 | (13.8–14.9) | 1.3 | (1.2–1.3) | 27.7% | (26.6–28.8) |
| Low and middle income countries | | | | | | |
| 2000 | 429.7 | (386.1–479.8) | 3.6 | (3.2–4.0) | 4.4% | (3.9–4.9) |
| 2010 | 420.3 | (374.9–459.3) | 3.3 | (3.0–3.6) | 6.1% | (5.5–6.7) |
| 2019 | 389.6 | (353.1–431.3) | 3.0 | (2.8–3.4) | 7.4% | (6.7–8.2) |

UI, uncertainty interval.

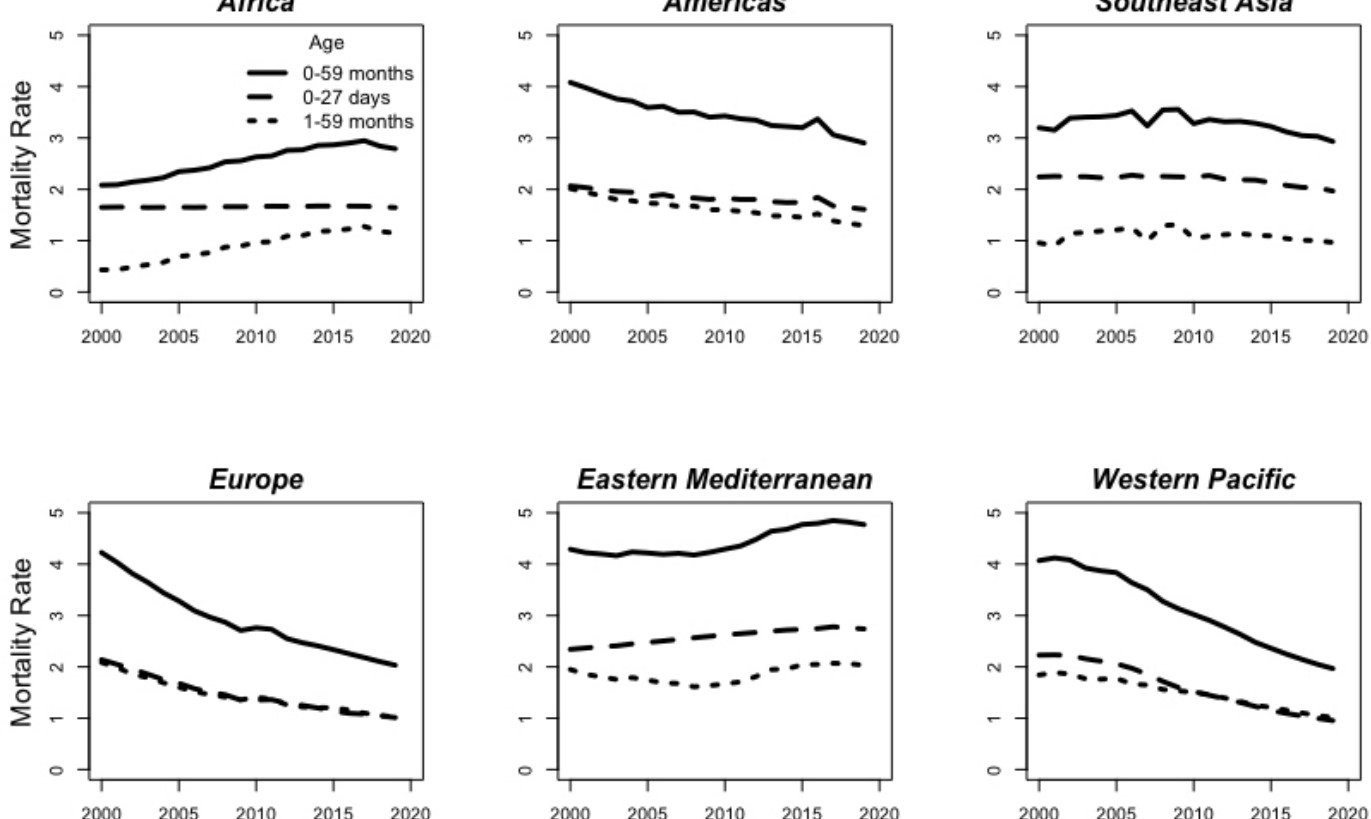

**Figure 1** Estimated rate of under-5 mortality due to birth defects in 2000–2019 by region. Mortality rate is shown per 1000 live births by age group.

Figure 1 depicts the estimated rate of under-5 mortality due to birth defects over time by WHO regional designations. The highest estimated regional rate in 2019 is in the Eastern Mediterranean region at 4.8 (95% UI 4.1–5.6), and the lowest is in the European region at 2.0 (95% UI 1.9–2.2). The rate of mortality due to birth defects is decreasing in some regions, notably Europe, the Americas and the Western Pacific, while the total rate of under-5 mortality due to birth defects is estimated to have risen in 2019 from 2000 in Africa and the Eastern Mediterranean. Figure 2 examines the estimated per cent of mortality due to birth defects over time by region and shows that the per cent of under-5 deaths due to birth defects is increasing in all age groups across all WHO regions. This per cent of mortality due to birth defects is not necessarily smooth over time due to extreme events such as the earthquake in Haiti (2010), affecting mortality in the Americas, as well as natural disasters in other areas. In 2019, Europe and the Americas regions have the highest per cent of mortality due to birth defects at 25.2% (95% UI 23.9%–27.4%) and 21.8% (95% UI 20.5%–23.5%), respectively, while the Africa region has the lowest per cent due to birth defects at 3.7% (95% UI 3.2%–4.3%).

A snapshot of overall country contributions to the per cent of under-5 deaths due to birth defects in 2019 is depicted in figure 3. Data represent 194 countries; each dot represents a country, delineated by high-income

countries and LMICs. The per cent of under-5 mortality due to birth defects is highly variable at low mortality rates, likely due to a low number of under-5 deaths in some countries. More generally, the per cent of mortality due to birth defects increases dramatically as under-5 mortality declines. In areas where under-5 mortality is greater than 25 per 1000 livebirths, only 5.4% of deaths are due to birth defects, whereas 22.8% of under-5 mortality is due to birth defects in countries with an under-5 mortality less than 25 per 1000 livebirths.

## DISCUSSION

The actual trends in under-5 mortality due to birth defects (rather than the estimated trends) are driven by two primary factors: the prevalence at birth and care after birth. Prevalence at birth is impacted by exposure to risk factors and preventive measures, preconception care and genetic counselling and availability and uptake of screening during pregnancy, including elective termination of pregnancies (TOP).[16] The care received after birth is influenced by diagnostic capacity of health systems as well as the availability of appropriate surgical and medical care and follow-up. Together, the complex interplay of these factors determines the real trend in under-5 mortality due to birth defects, while the trend estimated by MCEE may be affected by other factors. Data quality, in particular, may be evolving over time, especially

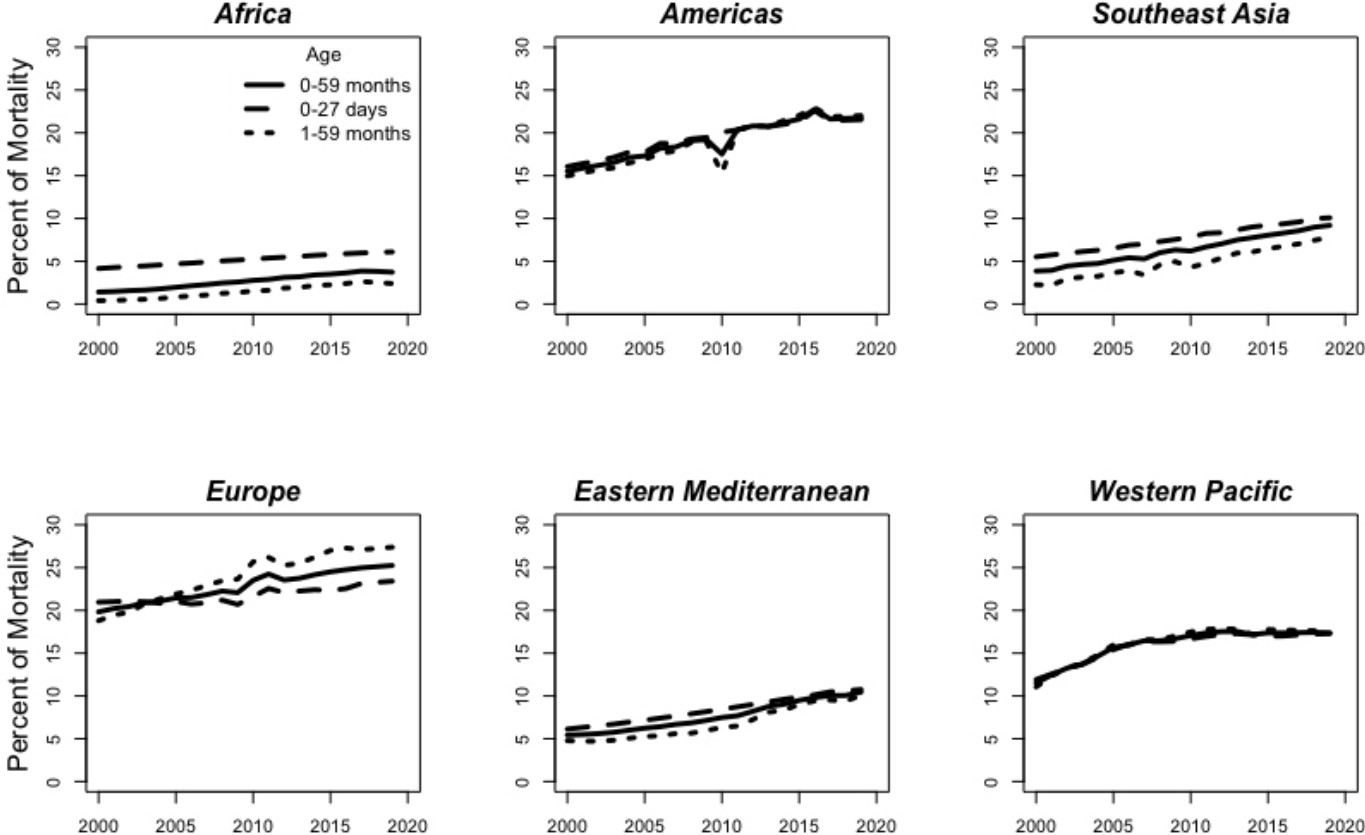

**Figure 2** Estimated per cent of mortality due to birth defects in 2000–2019 by region and age group.

in regions with countries that have sparse primary data such as Africa and Southeast Asia. For example, MCEE estimated an increase in the under-5 mortality rate due to birth defects in Africa and the Eastern Mediterranean since 2000 (figure 1), which may be due to increasing sensitivity to ascertaining birth defects in source data rather than an increase in actual mortality rates.

Despite the limitations, the MCEE estimates suggest that under-5 deaths due to birth defects are in decline, paralleling the overall decline in under-5 mortality. As advances

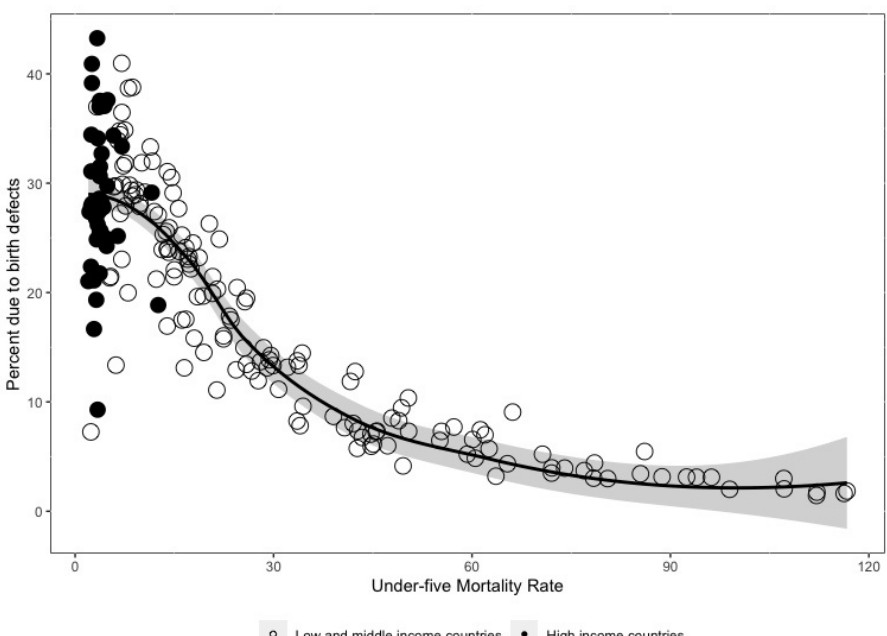

**Figure 3** Per cent of under-5 mortality due to birth defects by under-5 mortality rate among high and low income countries in 2019. Each dot represents a country, while the solid line is the local average smoothed across mortality rate.

in child survival are made globally and other underlying drivers of under-5 mortality are addressed (eg, infectious diseases of childhood), MCEE estimates suggest that mortality due to birth defects is assuming a larger proporation of under-5 mortality, as indicated by the increasing global per cent of under-5 deaths due to birth defects from 2000 to 2019. In 2000, birth defects were estimated to be the seventh highest cause of under-5 mortality globally, while in 2019, they were fifth. This increase in the proportion of under-5 deaths due to birth defects over time is largely attributable to reductions in deaths due to infectious causes such as diarrhoea, pneumonia, malaria, measles, HIV, sepsis and meningitis.[2] However, under-5 mortality due to prematuriy and intrapartum events has also declined at a faster rate than mortality due to birth defects since 2000, perhaps indicating that these conditions are more easily managed.[2] The increase over time in the proportion of under-5 mortality due to birth deaths may be under or overestimated, as the sensitivity of verbal autopsies can change depending on what other causes of morality are present.[17]

Improvements in under-5 survival over the past 20 years are demonstrated by declining mortality rates in both high-income countries and LMICs, although estimated mortality due to birth defects declined more slowly in general in LMICs and is not declining in all regions. Estimated trends may be sensitive to the quality of source information and changes in data quality over time as well as choices made in the estimation process. Some actual decrease over time in the mortality due to birth defects is also likely reflected in the MCEE estimates. In general, health systems have made improvements since 2000,[18] including advances in the management of birth defects such as congenital heart and neural tube defect repair, despite the ongoing challenges to manage these conditions in LMICs.[19] The implementation of birth defects prevention measures has also improved globally, for example, with folic acid supplementation and fortification.[20]

While the overall proportion of deaths due to birth defects is much higher in high-income countries, the absolute burden of under-5 mortality due to birth defects disproportionately impacts LMICs because the death rate from birth defects is more than two times as high than in high-income countries. The burden of birth defects relative to other causes of under-5 mortality in LMICs is also likely to increase as infectious causes such as acute respiratory infections and diarrheal diseases decline. High-income countries in general have much lower rates of mortality due to infectious diseases, contributing to their higher percentage of under-5 deaths being due to birth defects. Estimated mortality rates from birth defects were lower in high-income countries compared with LMICs (1.29 vs 3.03 deaths per 1000 live births). This difference is likely underestimated, given that high-income countries generally have their own vital registration data, while many estimates for LMICs are derived from verbal autopsy studies.

There are also differences in regional burden estimates that are likely due to the limitations of MCEE estimates derived from verbal autopsy studies. In 2000, the estimated mortality rate due to birth defects was similar in Europe, the Americas, the Western Pacific and the Eastern Mediterranean regions (4.2, 4.1, 4.1 and 4.3 per 1000 live births, respectively) but lower in Southeast Asia and Africa (3.2 and 2.1 per 1000 live births), despite the true birth prevalence likely being similar across regions[14] and a likely higher quality of care for birth defects in Europe and the Americas.[16] Together with the known limitations of verbal autopsy studies, this strongly suggests that deaths due to birth defects in Africa and Southeast Asia are underestimated.

The MCEE estimates should be considered the minimum lowest estimate for deaths attributable to birth defects and should not be used for estimating total burden of under-5 mortality attributable to birth defects in countries with modelled estimates. Verbal autopsy studies are used by the MCEE to derive estimates of cause-specific mortality in high under-5 all-cause mortality areas, and verbal autopsies in general have several important limitations related to birth defects. A verbal autopsy is a structured interview between a trained interviewer and a caregiver to the deceased, who generally are not trained to recognise birth defects among children or newborns apart from externally visible or otherwise obvious anomalies.[21–23] The MCEE estimates are also modelled conditional on a selection of covariates, such as under-5 mortality rate, total fertility rate and skilled birth attendance, and so some country estimates are partially driven by modelling choices.[2] The MCEE estimates for under-5s also do not include all deaths due to birth defects because neither stillbirths nor deaths among older children are included. MCEE does estimate mortality due to birth defects among children and adolescents 5–19 years olds, but those estimates were not examined here.[24]

Relative to causes of deaths reported from verbal autopsies, causes reported by vital registration systems are more standardised across countries, because they are based on medical certification of deaths and mapped to ICD codes. However, vital registration systems may underestimate mortality due to birth defects in some cases.[25] In addition, the ICD coding scheme has changed over time, and there is evidence that the prevalence of birth defects may be different when coded with ICD-10 compared with ICD-9, although previously observed effects varied by type of birth defect.[5] Countries are currently shifting to ICD-11 coding, which has expanded the number of codes for birth defects and might aid with standardisation between countries.[26] Systematic estimates for the total burden of under-5 mortality due to birth defects should provide estimates of burden at the national level in a comparable and systematic way over time, giving countries the opportunity to improve health system planning for current and future expected populations.

The MCEE estimates are developed separately from estimates related to the burden of birth defects estimated

by other groups, including those by the GBD project.[27] Unlike the MCEE estimates, those developed by GBD have not been formally reviewed by countries and, therefore, do not have an explicit aim of transparency.[2] The GBD estimates include a comprehensive series of both mortality and morbidity due to birth defects as well as estimates for specific types of birth defects, despite the sparseness of data in many LMICs.[28] Globally, the GBD estimates indicate a higher per cent of under-5 deaths due to birth defects in 2019 (9.4%) than the MCEE estimates (7.6%); however, the 2019 GBD estimates for deaths due to birth defects in many LMICs are a lower per cent of under-5 deaths than MCEE estimates, including some high burden countries such as Pakistan, Egypt, Syria and Venezuela (not shown). Similar to MCEE, the GBD estimates of mortality burden due to birth defects do not include stillbirths or TOP.[29] A systematic comparison between the WHO estimates of mortality due to birth defects and those produced by GBD is outside the scope of this research.

## Limitations

Our examination of the WHO estimates of under-5 mortality is limited. Our analysis did not include an examination of specific types of birth defects, as MCEE estimates are not currently produced for cause-specific mortality at a high enough resolution for such comparison. In addition, although other estimates for the mortality due to birth defects are available, notably from GBD, we have not systematically compared WHO/MCEE estimates to those of GBD, or to earlier estimates from the WHO and the March of Dimes.

## Conclusion

The MCEE has developed a useful set of estimates related to the burden of mortality among children under-5 related to birth defects, suggesting decreases in the rate of under-5 mortality due to birth defects since 2000 and increases in the per cent of under-5 mortality due to birth defects. However, both the MCEE-estimated rate and per cent of under-5 mortality due to birth defects are underestimated and represent only a minimum. More comprehensive estimates of this burden are needed for the most effective health system planning. The ideal estimates should encompass all deaths due to birth defects, including stillbirths and children older than 5, and they should not be subjected to the limitations of verbal autopsies. The best estimates would also have input from LMICs who are screening for and diagnosing birth defects and who may benefit from technical guidance in building capacity in this area. These estimates should also be developed to cover the recent time period, employ the latest methods to maximise transparency and replicability and incorporate the most recent data available from surveillance systems consistently across areas. Surveillance systems and estimates of the mortality burden of birth defects should not rely on verbal autopsies. Such estimates are a critical contribution to national, regional and global planning as the population affected by birth defects becomes a more prominent mortality burden among children under 5 years.

**Author affiliations**
[1]Department of International Health, Johns Hopkins University Bloomberg School of Public Health, Baltimore, Maryland, USA
[2]National Center on Birth Defects and Developmental Disabilities, Centers for Disease Control and Prevention, Atlanta, Georgia, USA
[3]Department of Maternal, Newborn, Child and Adolescent Health, and Ageing, World Health Organization, Geneve, Switzerland
[4]Department of Maternal, Newborn, Child and Adolescent Health, World Health Organization, Genève, Switzerland
[5]Department of Epidemiology and Population Health, London School of Hygiene & Tropical Medicine, London, UK
[6]Population, Family, and Reproductive Health, Johns Hopkins University Bloomberg School of Public Health, Baltimore, Maryland, USA

**Acknowledgements** The findings and conclusions in this report are those of the authors and do not necessarily represent the official position of the Centers for Disease Control and Prevention. The authors alone are responsible for the views expressed in this article and they do not necessarily represent the views, decisions or policies of the institutions with which they are affiliated.

**Contributors** TD, KLS, ADC, RJB and HB conceptualised the research plan, LL, JP and CTM designed the analysis. JP, CTM and JLW worked together on the original draft. All authors reviewed and revised the manuscript. JP accepts full responsibility for this research, had access to the data, and made the decision to publish.

**Funding** This research was supported in part by the Bill and Melinda Gates Foundation (INV-006966). The funders of the study did not have a role in this study design, data collection, data analysis, interpretation or writing of this manuscript.

**Competing interests** None declared.

**Patient and public involvement** Patients and/or the public were not involved in the design, or conduct, or reporting, or dissemination plans of this research.

**Patient consent for publication** Not applicable.

**Provenance and peer review** Not commissioned; externally peer reviewed.

**Data availability statement** Data are available in a public, open access repository. This research is based on estimates produced by the World Health Organization which are included as supplementary material and are also publicly available at https://github.com/amulick/MCEE-u5mort2019.

**ORCID iDs**
Jamie Perin http://orcid.org/0000-0002-5482-6620
Cara T Mai http://orcid.org/0000-0002-7300-0672
Ayesha De Costa http://orcid.org/0000-0003-1869-5990
Kathleen Strong http://orcid.org/0000-0002-8092-1955
Theresa Diaz http://orcid.org/0000-0001-5063-8078
Hannah Blencowe http://orcid.org/0000-0003-1556-3159
Robert J Berry http://orcid.org/0000-0002-7162-5046
Jennifer L Williams http://orcid.org/0000-0002-1716-7005

Li Liu http://orcid.org/0000-0003-3102-6708

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
