## [Reviewer comments · BMJ Open]

ARTICLE DETAILS

TITLE (PROVISIONAL)	Systematic estimates of the global, regional and national under-five mortality burden attributable to birth defects in 2000-2019: a summary of findings from the 2020 WHO estimates
AUTHORS	Perin, J; Mai, Cara T; De Costa, Ayesha; Strong, Kathleen Louise; Diaz, Theresa; Blencowe, Hannah; Berry, Robert; Williams, Jennifer; Liu, Li

VERSION 1 – REVIEW

REVIEWER	Wojcik, Monica Boston Children's Hospital
REVIEW RETURNED	05-Sep-2022

GENERAL COMMENTS	This manuscript addresses an important issue related to the global child mortality burden. The methods are clearly described and defined and the data are also clearly presented. The discussion is also well-structured, leading with the complex interplay between the number of pregnancies affected by congenital anomalies that proceed to live birth in addition to care after birth in determining outcomes. The authors address limitations in data reporting that may lead to both underreporting of such deaths or an apparent increase in mortality due to improved ascertainment. Overall, I think that this manuscript presents helpful data towards understanding of a complex and important public health issue. I did wonder if there were any differences noted among specific birth defects or classes of birth defects (i.e. neural tube, cardiac anomalies) that may be informative in understanding the limitations of the authors' techniques? For example, do the limitations of verbal autopsy lead to skewing of mortality reports in certain types of anomalies?
--

REVIEWER	Poenaru, Dan MyungSung Christian Medical Center, Pediatric Surgery
REVIEW RETURNED	20-Oct-2022

GENERAL COMMENTS	This is an analysis of 20-year trends in under-5 mortality and specific mortality from birth defects based on one specific group, the WHO MCEE. The manuscript is solid, well written, and thorough in its methodology and reporting of results. I cannot comment on the correctness of the statistical analyses. My comments and suggestions are as follows: While the paper title gives the impression of an overall, authoritative, 20-year analysis of birth defects deaths, and the Introduction clearly lists several agencies generating such parallel
---

	data, the current study limits itself to one source (MCEE). This single-source analysis and critique must therefore be made clear in the title and in the abstract (which doesn't even mention MCEE as the data source). In the same vein, if the authors' critique of the MCEE data (the poor quality of verbal autopsy data) is of limited use in the absence of a more detailed comparison of the MCEE data and data sources with all other existing birth defect registries. Without such side-by-side comparisons, any assertions regarding the need for other methods remain just that - subjective assertions. The Limitations section is notably absent. Finally, the points above ultimately result in a solid analysis with a very weak and limited message, and questionable applicability and generalizability. If the data sources used by MCEE are unreliable (as the authors posit), then what can we take home as conclusions regarding the results themselves?
--	---

REVIEWER	Anderson, Jamie University of California Davis
REVIEW RETURNED	25-Oct-2022

GENERAL COMMENTS	This paper estimates the burden of disease of congenital anomalies as well as the burden of mortality as a result of congenital anomalies. It makes these estimates from vital registration systems and verbal autopsy studies. Methods: Can you explain how the verbal autopsies are reported? Are they specific as to which defect the baby has? Is there any way to track this? Is there a standard methodology for verbal autopsies? You briefly describe this in the discussion but how they actually identify birth defects would be helpful in the methods. Is there any way to identify which congenital anomalies are impacting the overall rates, especially by region and over time? Even broad categories would be helpful, such as cardiac anomalies, gastrointestinal, multiple congenital anomalies, etc. I assume that prematurity is not considered a congenital anomaly, but would like to confirm. Are premature babies with comorbid congenital anomalies theoretically counted in this dataset? I'd recommend expanding the discussion on why these percentages have changed over time. You mention infectious diseases, but what other changes to childhood mortality have affected the relative mortality rate by congenital birth defects? I realize this is broad and varies by region, but maybe this would help explain some regional differences. Overall, while there are significant limitations, many of which the authors do address, it is at least an estimate of the overall burden of disease of congenital anomalies and is thus worthwhile for publication.
--

VERSION 1 – AUTHOR RESPONSE

Reviewer: 1

Dr. Monica Wojcik, Boston Children's Hospital

Comments to the Author:

This manuscript addresses an important issue related to the global child mortality burden. The methods are clearly described and defined and the data are also clearly presented. The discussion is also well-structured, leading with the complex interplay between the number of pregnancies affected by congenital anomalies that proceed to live birth in addition to care after birth in determining outcomes. The authors address limitations in data reporting that may lead to both underreporting of such deaths or an apparent increase in mortality due to improved ascertainment.

Overall, I think that this manuscript presents helpful data towards understanding of a complex and important public health issue. I did wonder if there were any differences noted among specific birth defects or classes of birth defects (i.e. neural tube, cardiac anomalies) that may be informative in understanding the limitations of the authors' techniques? For example, do the limitations of verbal autopsy lead to skewing of mortality reports in certain types of anomalies?

Response: It is generally thought that verbal autopsies are more suited for reporting the types of birth defects that are easily observed among people without medical training (for example, oro-facial cleft) compared to birth defects that need special diagnostic tools (such as congenital heart disease). However, the WHO/MCEE has not produced estimates of mortality due to birth defects by type of birth defect, so we have not examined estimates related to this. Given that the WHO estimates appear biased for the total mortality due to birth defects, it is unlikely that many specific types of birth defect mortality could be estimated well with similar methods.

Reviewer: 2

Dr. Dan Poenaru, MyungSung Christian Medical Center, Montreal Children's Hospital Research Institute

Comments to the Author:

This is an analysis of 20-year trends in under-5 mortality and specific mortality from birth defects based on one specific group, the WHO MCEE. The manuscript is solid, well written, and thorough in its methodology and reporting of results. I cannot comment on the correctness of the statistical analyses.

My comments and suggestions are as follows:

While the paper title gives the impression of an overall, authoritative, 20-year analysis of birth defects deaths, and the Introduction clearly lists several agencies generating such parallel data, the current study limits itself to one source (MCEE). This single-source analysis and critique must therefore be made clear in the title and in the abstract (which doesn't even mention MCEE as the data source).

Response: We have edited the title and abstract to reflect that we have only examined a single source for the estimates of under-five mortality due to birth defects.

In the same vein, if the authors' critique of the MCEE data (the poor quality of verbal autopsy data) is of limited use in the absence of a more detailed comparison of the MCEE data and data sources with all other existing birth defect registries. Without such side-by-side comparisons, any assertions regarding the need for other methods remain just that - subjective assertions.

Response: The GBD estimates of cause-specific mortality are not developed with the objective of transparency and replicability and are not reviewed by countries in the same way as the WHO/MCEE estimates. The WHO/MCEE are the only estimates in this class of transparent estimates that are developed for all countries in a comprehensive time series. Despite this drawback of the GBD estimates, we agree that a comprehensive systematic comparison between WHO and GBD would be useful to inform future estimation processes. Toward this end, WHO has formed an advisory group to update the Birth Defect mortality and morbidity estimates which will include a review of GBD along with earlier estimates of the March of Dimes. However, such an analysis was beyond the scope or focus of this current research. We have added a comment to acknowledge this limitation, adding:

A systematic comparison between the WHO estimates of mortality due to birth defects and those produced by GBD is outside the scope of this research.

The Limitations section is notably absent.

Response: We have included a section for limitations of our study, adding the text below:

Our examination of the WHO estimates of under-five mortality is limited. Our analysis did not include an examination of specific types of birth defects, as MCEE estimates are not currently produced for cause-specific mortality at a high enough resolution for such comparison. In addition, although other estimates for the mortality due to birth defects are available, notably from GBD, we have not systematically compared WHO/MCEE estimates to those of GBD, or to earlier estimates from the WHO and the March of Dimes.

Finally, the points above ultimately result in a solid analysis with a very weak and limited message, and questionable applicability and generalizability. If the data sources used by MCEE are unreliable (as the authors posit), then what can we take home as conclusions regarding the results themselves?

Response: This analysis is meant to be a systematic assessment of what is currently estimated by the WHO for under-five mortality due to birth defects, describing their limitations and how they can be interpreted as they currently stand, it is not meant to present these estimates of mortality as the ground truth. We have clarified our objective and intentions with this research, related to several above comments.

Reviewer: 3

Dr. Jamie Anderson, University of California Davis

Comments to the Author:

This paper estimates the burden of disease of congenital anomalies as well as the burden of mortality as a result of congenital anomalies. It makes these estimates from vital registration systems and verbal autopsy studies.

Methods: Can you explain how the verbal autopsies are reported? Are they specific as to which defect the baby has? Is there any way to track this? Is there a standard methodology for verbal autopsies? You briefly describe this in the discussion but how they actually identify birth defects would be helpful in the methods.

Response: Verbal autopsies are interviews with caregivers of the deceased, which are summarized to estimate the cause of death with an algorithm. The interviews themselves are standardized, and there are several algorithms currently being used to determine cause of death, including physician review, InterVA, and InSilicoVA. To our knowledge these algorithms have not been examined for specific birth defects among under-five deaths. We have included the highlighted text below in the methods section:

The cause of these deaths is determined with the best available information, which is obtained from a verbal autopsy interview with family or caregivers which are then summarized to determine cause of death with an algorithm, several of which are in common use. Verbal autopsy interviews generally include at least one question related to birth defects, such as “Was any part of the baby physically abnormal at the time of delivery? (for example: body part too large or too small, additional growth on body)“, which can be utilized by a reviewing physician or algorithm to determine mortality due to birth defects.

Is there any way to identify which congenital anomalies are impacting the overall rates, especially by region and over time? Even broad categories would be helpful, such as cardiac anomalies, gastrointestinal, multiple congenital anomalies, etc.

Response: The WHO has not produced estimates for mortality due to specific types of birth defects. It is questionable whether there is adequate information to create these estimates. These estimates are not within the scope of this research.

I assume that prematurity is not considered a congenital anomaly, but would like to confirm. Are premature babies with comorbid congenital anomalies theoretically counted in this dataset?

Response: WHO/MCEE have produced estimates for mortality due to prematurity, which is not here considered a congenital anomaly. This analysis includes estimates of cause-specific mortality where the cause is determined by ICD-10 or ICD-11 rules (in countries with vital registration), where mortality among infants and children with evidence of both prematurity and birth defects are generally attributed to birth defects, since birth defects earlier than premature birth in the series of events.

Reference

ICD 11 Reference Guide. <https://icdcdn.who.int/icd11referenceguide/en/html/index.html>

I'd recommend expanding the discussion on why these percentages have changed over time. You mention infectious diseases, but what other changes to childhood mortality have affected the relative mortality rate by congenital birth defects? I realize this is broad and varies by region, but maybe this would help explain some regional differences.

Response: We have included some additional discussion related to the change in the proportion of under-five deaths due to birth defect over time, adding:

This increase in the proportion of under-five deaths due to birth defects over time is largely attributable to reductions in deaths due to infectious causes such as diarrhea, pneumonia, malaria, measles, HIV, sepsis, and meningitis. However, under-five mortality due to prematurity and intrapartum events has also declined at a faster rate than mortality due to birth defects since 2000, perhaps indicating that these conditions are more easily managed.

VERSION 2 – REVIEW

REVIEWER	Poenaru, Dan MyungSung Christian Medical Center, Pediatric Surgery
REVIEW RETURNED	21-Dec-2022

GENERAL COMMENTS	My concerns have been adequately addressed.
---

REVIEWER	Anderson, Jamie University of California Davis
REVIEW RETURNED	09-Jan-2023

GENERAL COMMENTS	The edits made by the authors are acceptable and I recommend publication of the article.
--